# Is MRPI 2.0 More Useful than MRPI and M/P Ratio in Differential Diagnosis of PSP-P with Other Atypical Parkinsonisms?

**DOI:** 10.3390/jcm11102701

**Published:** 2022-05-10

**Authors:** Natalia Madetko, Piotr Alster, Michał Kutyłowski, Bartosz Migda, Michał Nieciecki, Dariusz Koziorowski, Leszek Królicki

**Affiliations:** 1Department of Neurology, Medical University of Warsaw, 03-242 Warsaw, Poland; dariusz.koziorowski@wum.edu.pl; 2Department of Radiology, Mazovian Brodnowski Hospital, 03-242 Warsaw, Poland; michael.kutylowski@gmail.com; 3Diagnostic Ultrasound Lab, Department of Pediatric Radiology, Medical Faculty, Medical University of Warsaw, 03-242 Warsaw, Poland; bartoszmigda@gmail.com; 4Department of Nuclear Medicine, Children’s Memorial Health Institute, 04-730 Warsaw, Poland; msnieciecki@gmail.com; 5Department of Nuclear Medicine, Mazovian Brodno Hospital, 03-242 Warsaw, Poland; leszek.krolicki@wum.edu.pl; 6Department of Nuclear Medicine, Medical University of Warsaw, 02-097 Warsaw, Poland

**Keywords:** PSP-P, MRI, MRPI, progressive supranuclear palsy, neuroimaging

## Abstract

Differential diagnosis of progressive supranuclear palsy remains difficult, especially when it comes to the parkinsonism predominant type (PSP-P), which has a more favorable clinical course. In this entity, especially during the advanced stages, significant clinical overlaps with other tauopathic parkinsonian syndromes and multiple system atrophy (MSA) can be observed. Among the available additional diagnostic methods in every-day use, magnetic resonance imaging (MRI) focused specifically on the evaluation of the mesencephalon seems to be crucial as it is described as a parameter associated with PSP. There is growing interest in relation to more advanced mesencephalic parameters, such as the magnetic resonance parkinsonism index (MRPI) and MRPI 2.0. Based on the evaluation of 74 patients, we demonstrate that only the mesencephalon/pons ratio and MRPI show a significant difference between PSP-P and MSA-parkinsonian type (MSA-P). Interestingly, this differential feature was not maintained by MRPI 2.0. The mesencephalon to pons ratio (M/P), MRPI and MRPI 2.0 were not found to be feasible for the differentiation of PSP-P from other atypical tauopathic syndromes.

## 1. Introduction

Progressive supranuclear palsy-parkinsonism predominant (PSP-P) is the second most common phenotype of PSP [1]. It is characterized by a more favorable clinical course than PSP-Richardson syndrome (PSP-RS). However, the examination of PSP-P can be problematic due to an overlapping clinical manifestation with Parkinson’s disease (PD) and other atypical parkinsonian syndromes. Possible dysautonomia and less pronounced cognitive deterioration are examples of symptoms commonly associated with multiple system atrophy-parkinsonian type (MSA-P) that can be present in PSP-P [2,3]. Neuroimaging is seen as a potential supplementary method for examining atypical parkinsonisms [2,3,4]. In PSP-P, the factors best assessed by neuroimaging have yet to be fully explored. MRI parameters are commonly used to perform differential diagnosis of PSP-P and PD in the early stages, but less is known in the context of the comparative analysis of PSP-P and MSA [5,6]. Among the features commonly associated with PSP-P are the evaluation of magnetic resonance parkinsonism index (MRPI) and its upgraded edition, MRPI 2.0. The assessment of atypical parkinsonisms using easily accessible methods is impacted by their limited specificity. The goal of this research was to verify whether assessment using different parameters generally based on the evaluation of the mesencephalon is beneficial.

## 2. Material and Methods

The authors examined 75 patients (19 patients—12 males, 7 females—with PSP-RS; 16 patients—6 males, 10 females—with PSP-P; 19 patients—1 male, 18 females—with corticobasal syndrome (CBS); 21 patients with MSA-P—8 males, 13 females) as well as 16 controls (Table 1). All of the diagnoses were based on the most contemporary criteria for diagnosis [2,3,4]. The duration of disease varied from 3 to 6 years. The control group consisted of 9 females and 7 males aged from 52 to 88 diagnosed with headache, vertigo or epilepsy without any significant changes in neuroimaging. Except for hypertension, headaches and epilepsy, no clinically significant comorbidities were observed in the control group. All of the patients were examined in the Department of Neurology of the Medical University of Warsaw between January 2017 and December 2019. The group of controls was based on age-matched healthy volunteers.

### 2.1. Magnetic Resonance Imaging

Standard protocol magnetic resonance imaging using a 3.0 Tesla Magnetic Resonance (MR) Siemens Skyra system was performed on all of the patients included in the study. The images were evaluated, and all of the measurements used in the study were obtained by radiologists with at least 5 years of experience in neuroimaging. For the calculation of the average width of the third ventricle (V3), three measurements in the axial plane T2-weighted sequence at the level of the anterior and posterior commissure were acquired. The maximal left to right width of the frontal horns of the lateral ventricles (FH) was obtained in the same sequence parallel to this plane. The area of the pons (P) and the midbrain (M) were determined using Siemens Syngo.via workstation tools in the midsagittal plane of the T2-weighted sequence. For the measurement of the widths of the middle cerebellar peduncle (MCP) and superior cerebellar peduncle (SCP), the T2-weighted sagittal and coronal planes were used, respectively. The MRPI was calculated using the formula MRPI = (P/M) × (MCP/SCP). To obtain the value of MR parkinsonism index 2.0 (MRPI 2.0), the MRPI was multiplied by the ratio of the average width of the third ventricle and the maximal width of the frontal horns of the lateral ventricles according to formula: MRPI 2.0 = (MRPI) × (V3/FH). Figure 1, Figure 2, Figure 3, Figure 4 and Figure 5 illustrate the above-mentioned parameters.

### 2.2. Statistical Analysis

All calculations were performed with Statistica software (version 13.1. Dell. Inc. Statsoft, Round Rock, TX, USA).

The data gathered were assessed with non-parametric tests due to their non-Gaussian distribution. We compared PSP-P to other atypical parkinsonian entities (PSP-RS, MSA-P and CBS) and the control group in relation to the following MRI parameters: width of III ventricle, midbrain surface, mesencephalon-to-pons ratio (M/P), MRPI and MRPI 2.0. All results were presented as means with minimal and maximal values followed by the standard deviation and 95% confidence interval. The Kruskal–Wallis ANOVA was used for group comparisons and for post-hoc analysis pairwise multiple-comparison of mean ranks (PMCMR).

In the final assessment, we performed receiver operating characteristics (ROC) curve analysis in relation to statistical significance results from the PMCMR calculations. As a result, we determined cut-off values (using the Youden index) for each parameter with an area under curve (AUC) value and its 95% CI, as well as values for sensitivity, specificity, positive predictive value (PPV), negative predictive value (NPV), accuracy and figure as appropriate.

In terms of multiple comparison correction, we performed a Bonferroni correction to control the false discovery rate (FDR). The calculated *p*-value was 0.002 and was used as a threshold for all calculations as significant.

## 3. Results

In Table 1, we have presented descriptive statistics for all MRI parameters, providing mean values with the range (minimal and maximal values) as well as the standard deviation with a 95% confidence interval. The results of our ROC analysis for M/P ratio, MRPI 2.0, MRPI and midbrain area are shown in Figure 6, Figure 7, Figure 8 and Figure 9, respectively.

Overall, in the subgroup analysis there were only statistically significant differences between the control group and the PSP-P and MSA-P groups in relation to midbrain surface (0.76 vs. 1.21 and 0.76 vs. 1.08), M/P ratio (0.16 vs. 0.245 and 0.16 vs. 0.239), MRPI (17.384 vs. 11.247 and 17.384 vs. 10.766) and MRPI 2.0 PSP-P vs. control (4.338 vs. 2.342), all *p* < 0.002, as shown in Table 2. Unfortunately, the new MRI parameter MRPI 2.0 was found to be insignificant in relation to the corrected p-value when we compared patients with PSP-P and MSA-P, since *p* was 0.0038 > 0.002. From the ROC curve analysis, the best differentiating parameter for PSP-P and MSA-P patients (highest AUC) was M/P ratio (cut off = 0.177) followed by midbrain area (cut off = 0.85) and MRPI (cut off = 12.846). M/P ratio and midbrain area were characterized by the same values for sensitivity, specificity, accuracy, PPN and NPV, but the AUC 95% CI for M/P ratio was narrower. Both parameters had the highest specificity, PPV and accuracy (0.905, 0.846 and 0.882, respectively). MRPI was characterized by the highest values for sensitivity (0.923) and NPV (0.929), as shown in Table 3.

Comparison of PSP-P with the control group revealed M/P ratio was the best parameter for differentiating both groups, since AUC = 0.995 with a cut-off value of 0.202 with very high values for sensitivity, specificity, accuracy, PPV and NPV (all above 0.9, Table 3). If we apply the same cut-off value as the MSA-P group (cut off = 0.177), sensitivity, NPV and accuracy are lower (0.846 vs. 1, 0.889 vs. 1 and 0.931 vs. 0.966, respectively), but specificity and PPV are higher (1 vs. 0.938 and 1 vs. 0.929, respectively).

The rest of the parameters also had very high AUC values ranked in decreasing order as follows: MRPI AUC = 0.971 (cut off = 12.846), midbrain area AUC = 0.938 (cut off = 1.03) and MRPI 2.0 AUC = 0.928 (cut off = 3.062). All the values for sensitivity, specificity, accuracy, PPV and NPV are listed in Table 3.

Finally, if we apply the same cut-off value for midbrain area for the control group as the MSA-P group (cut off = 0.85), we see similar changes in sensitivity (0.846 vs. 1), accuracy (0.862 vs. 0.897), NPV (0.875 vs. 1), specificity (0.875 vs. 0.813) and PPV (0.846 vs. 0.813) as the M/P ratio.

## 4. Discussion

The outcome of this study suggests that among patients with a disease duration between 3 and 6 years, the differentiation between PSP-P and other tauopathic parkinsonian syndromes cannot be based on parameters related to the atrophy of mesencephalon (M/P ratio, MRPI, MRPI 2.0) or dilatation of the third ventricle (MRPI 2.0) from MRI. Interestingly, only two (M/P ratio, MRPI) out of the three parameters evaluated showed significant differences between PSP-P and MSA-P. The fact that the parameter that additionally takes into account the width of the third ventricle was not significant is intriguing considering that in multiple studies, the width of the third ventricle was found to be a helpful feature in differentiating PSP from PD [7,8]. Various works have stressed the significance of using the width of the third ventricle in transcranial sonography to differentiate between PSP and CBS [9,10]. The assessment of the third ventricle was found to be a feature in the evolution of PSP [11].

The M/P ratio, although affected by aging (as indicated in a study examining PD and PSP [8]), seems to be a more helpful parameter compared to MRPI 2.0 for the differentiation between PSP-P and MSA-P. In a study evaluating PD and PSP, MRPI was found to be a less variable parameter due to aging [12]. In another work, MRPI 2.0 was introduced as a parameter and showed higher specificity and sensitivity than MRPI for differentiating between early-stage PSP-P and PD [13]. In a study published in 2008, MRPI was found to have 100% sensitivity and specificity in avoiding misdiagnosis of patients with PSP. The use of this factor was also found to be beneficial for the differential diagnosis of PD and MSA, however the authors did not examine subgroups of patients with PSP-RS and PSP [14]. In PSP-P, MRPI was found to be a predictor of vertical supranuclear gaze palsy [15]. In another work, the authors showed that MRPI should be considered as a reliable examination for predicting the evolution of clinically unclassified parkinsonism into PSP [16]. MRPI 2.0 has been found to be more feasible for evaluating PSP progression [17]. A more recent study showed certain limits to the usefulness of MRPI and MRPI 2.0 since significant overlaps were found with normal pressure hydrocephalus and PSP [18].

Midbrain atrophy, the basic feature of the M/P, MRPI and MRPI 2.0 parameters, is a relevant but not necessarily useful factor in the examination of PSP. Previous studies showed limitations of the significance of this parameter in the context of differentiating the two most common phenotypes of PSP—PSP-RS and PSP-P—in the advanced stages. The progress of midbrain atrophy differs between PSP-RS and PSP-P, however in the advanced stages, an assessment of the midbrain is more beneficial for the differentiation of PSP-P and PD [19]. A different work suggested the assessment of SCP (a parameter used in MRPI and MRPI 2.0), the more damaged structure in PSP-RS, as a possibly sensible examination for the differentiation of PSP-P and PSP-RS [20]. More detailed examinations concerning PSP-P and other less common variants of PSP revealed atrophies within the globus pallidus, frontal lobe, cerebral peduncle, superior cerebellar peduncle and cerebellar as potentially feasible assessments for the radiological evaluation of PSP. In the work by Sakurai et al., the authors reviewed the neuroimaging findings for various subtypes, e.g., PSP with clinical manifestation of CBS, PSP-pure akinesia gait freezing, PSP-cerebellar subtype [21]. Taking into account the fact that the work by Sakurai et al. was published before the release of recent criteria for the diagnosis of PSP, more research in this area based on larger groups will likely provide a broader perspective in the context of this issue [2].

This study is affected by certain limitations. All of the diagnoses were based on clinical manifestation. No neuropathological examinations were conducted. The classification of patients was based on possible or probable diagnosis. The number of patients examined in each group was relatively small, but this can be partly explained by the rarity of the syndromes. The group of patients with MSA-P could not be age-matched due to the earlier incidence of this entity among patients.

## 5. Conclusions

The assessment of the mesencephalon and parameters based on its atrophy may be useful in the differentiation of synucleinopathic and tauopathic parkinsonian syndromes, but they do not provide sufficient specificity for the differential diagnosis of the different tauopathic parkinsonian syndromes. The usefulness of M/P ratio, MRPI and MRPI 2.0, though widely evaluated for the differentiation of PSP-P and PD as well as PSP, MSA and PD, were not previously evaluated in the context of the broad differentiation of PSP-P and other atypical parkinsonisms. To the best of our knowledge, this is the first work to evaluate PSP-P in this context. Our work shows that MRPI 2.0 is less beneficial for the differential diagnosis of the two most problematic entities among the atypical parkinsonisms—MSA-P and PSP-P—than MRPI and M/P ratio. The fact that significant differences in the evaluation of M/P ratio and MRPI are only observed in the differentiation of PSP-P and MSA-P suggests the need for developing better tools for the differential diagnosis of PSP-P and other tauopathic parkinsonian syndromes. Due to the limitations mentioned in the discussion, this work should be interpreted as encouraging further discussion in the field rather than providing a definitive observation.

## Figures and Tables

**Figure 1 jcm-11-02701-f001:**
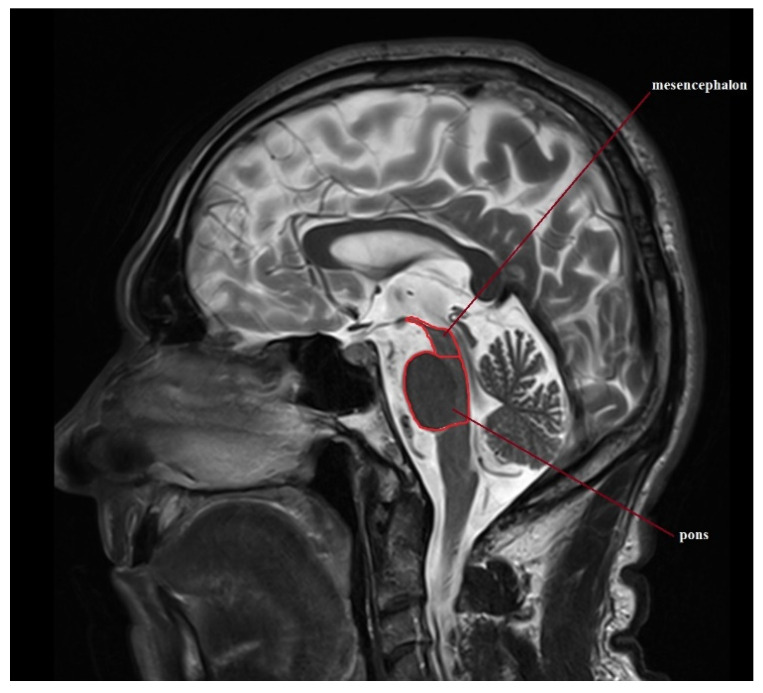
Mesencephalon-to-pons ratio (M/P ratio) for a patient with PSP-RS.

**Figure 2 jcm-11-02701-f002:**
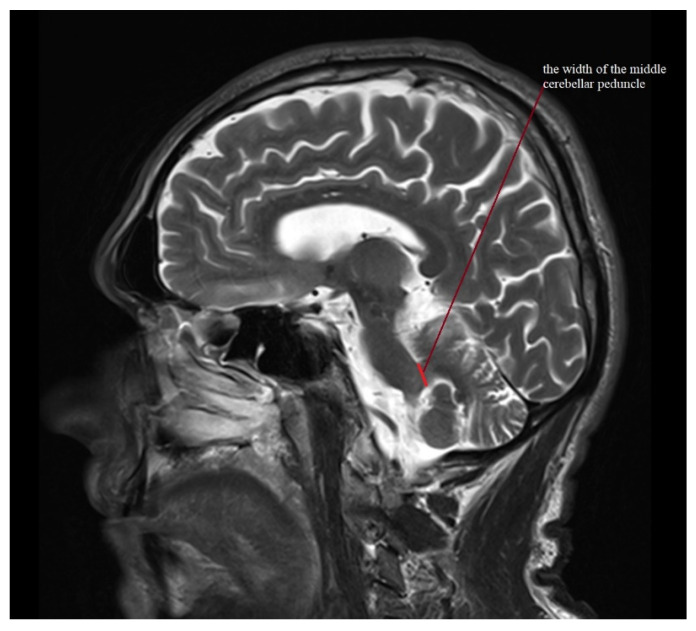
Measurement of the width of the middle cerebellar peduncle (MCP) for a patient with PSP-RS.

**Figure 3 jcm-11-02701-f003:**
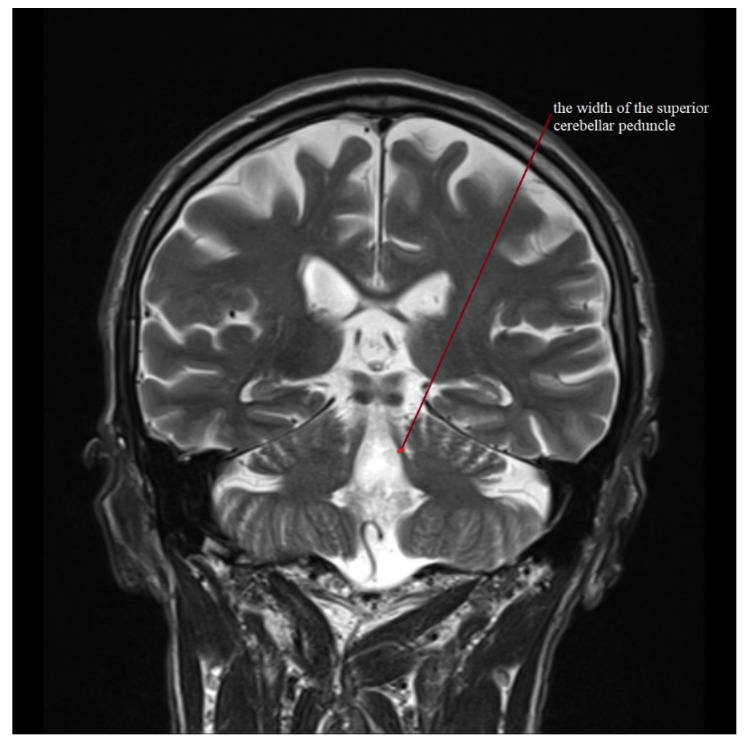
Measurement of the width of the superior cerebellar peduncle (SCP) for a patient with PSP-RS.

**Figure 4 jcm-11-02701-f004:**
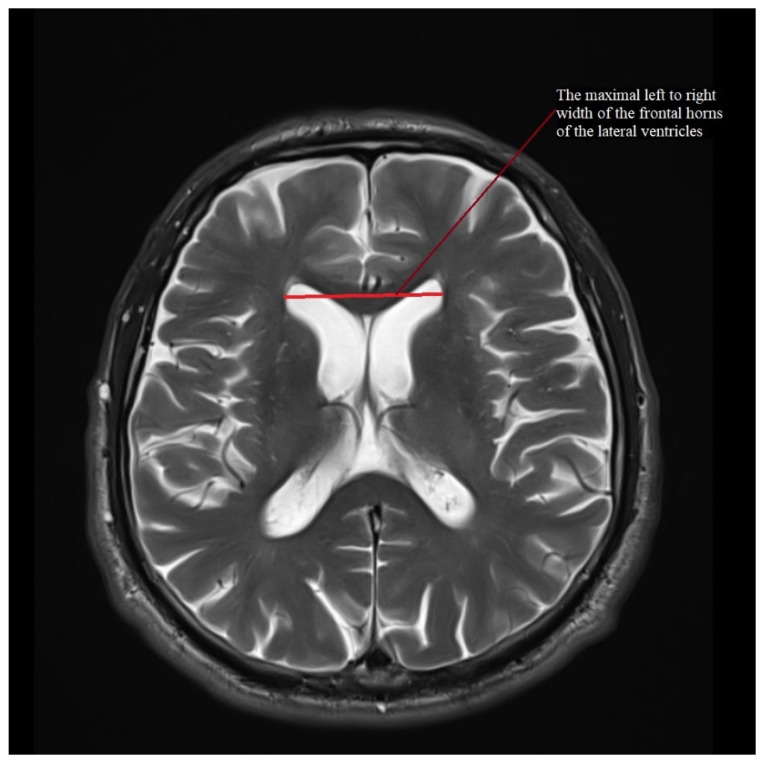
The maximal left to right width of the frontal horns of the lateral ventricles (FH) for a patient with PSP-RS.

**Figure 5 jcm-11-02701-f005:**
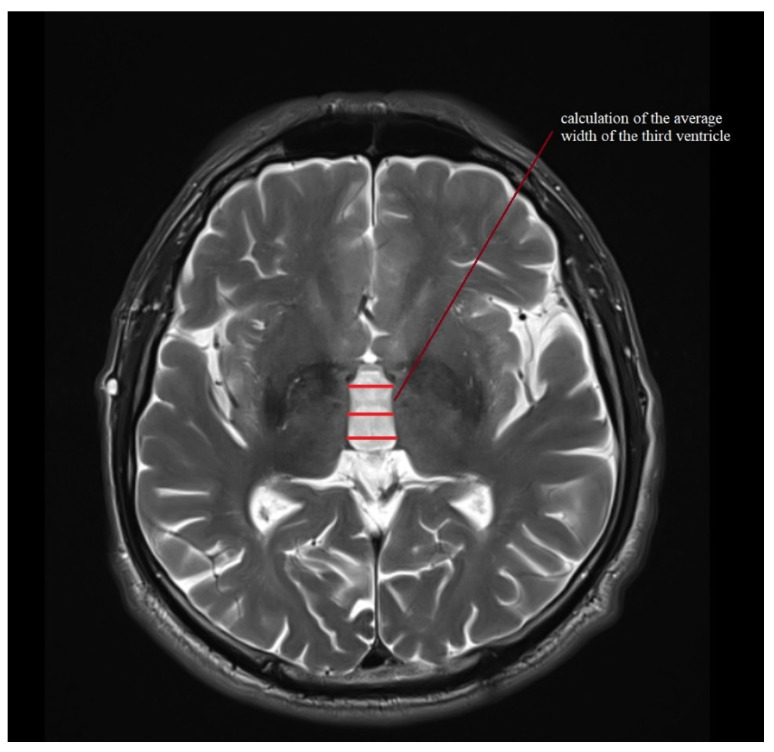
Calculation of the average width of the third ventricle (V3) for a patient with PSP-RS.

**Figure 6 jcm-11-02701-f006:**
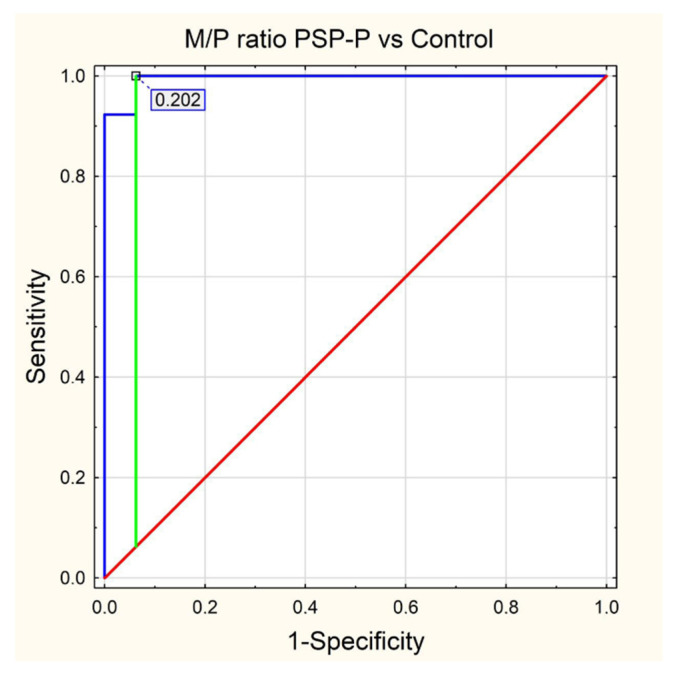
ROC curve—M/P ratio PSP-P vs. Control.

**Figure 7 jcm-11-02701-f007:**
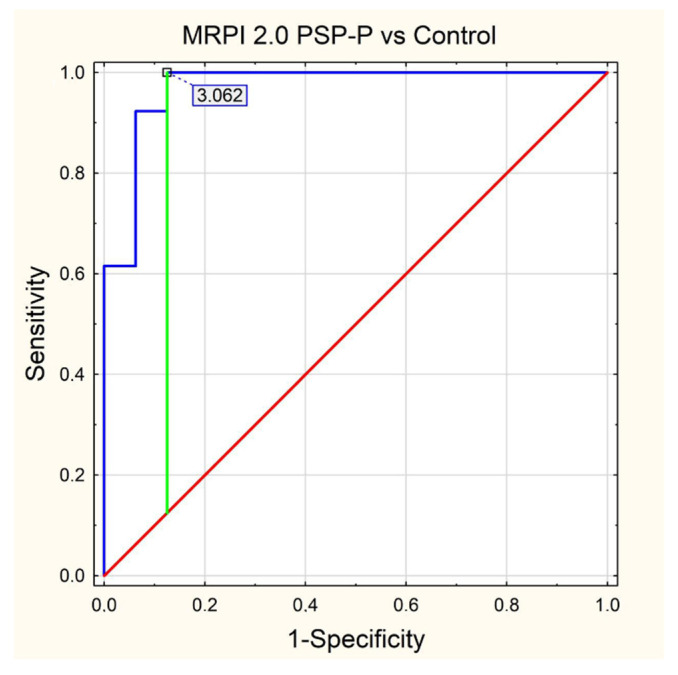
ROC curve—MRPI 2.0 PSP-P vs. Control.

**Figure 8 jcm-11-02701-f008:**
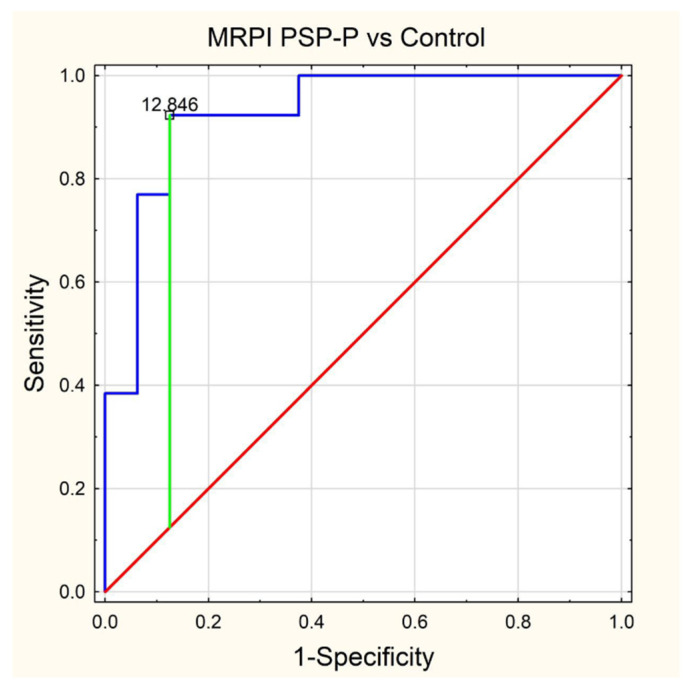
ROC curve—MRPI PSP-P vs. Control.

**Figure 9 jcm-11-02701-f009:**
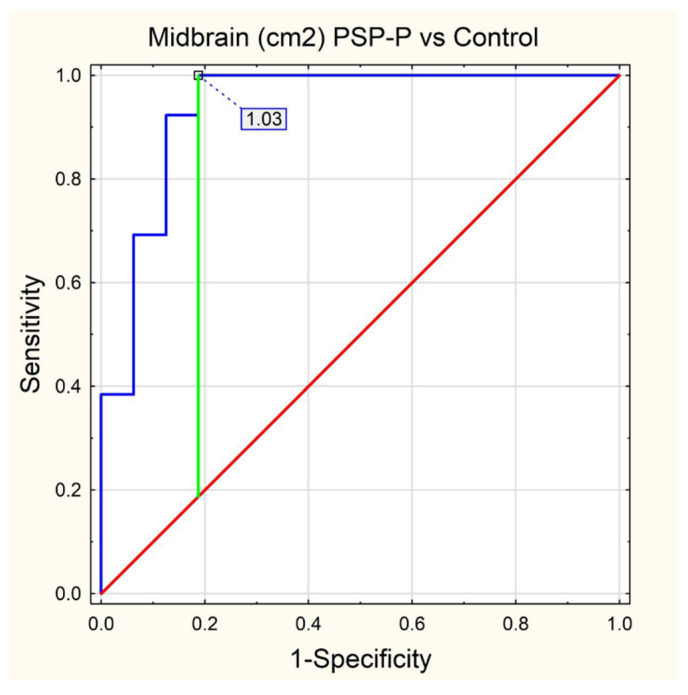
ROC curve—Midbrain (cm^2^) PSP-P vs. Control.

**Table 1 jcm-11-02701-t001:** Descriptive statistics.

	Control (N = 16)		PSP-P (N = 16)		PSP-RS (N = 19)		MSA-P (N = 21)		CBS (N = 19)	
	M/F = 7/9		M/F = 6/10		M/F = 12/7		M/F = 8/13		M/F = 1/18	
	Mean (Min–Max)	SD ± 95% CI	Mean (Min–Max)	SD ± 95% CI	Mean (Min–Max)	SD ± 95% CI	Mean (Min–Max)	SD ± 95% CI	Mean (Min–Max)	SD ± 95% CI
Age	68.9 (52–88)	10.2 ± 7.6–15.9	71.4 (61–81)	6.9 ± 5.1–10.6	74 (62–83)	5.8 ± 4.4–8.5	62.6 (50–81)	8.9 ± 6.8–12.8	72.8 (57–87)	7.2 ± 5.4–10.6
UPDRS III ON	32.6 (4–80)	17 ± 13.7–22.4	32.2 (16–63)	17.9 ± 12.1–34.2	36.5 (4–54)	17 ± 11.9–29.8	35 (20–80)	25.5 ± 15.3–73.1	26.9 (14–49)	11.4 ± 7.7–21.8
UPDRS III OFF	37.9 (11–87)	18.3 ± 14.8–24.1	38.3 (20–69)	17.9 ± 12.1–34.3	43.6 (11–72)	17.5 ± 12.2–30.7	37.8 (21–87)	27.8 ± 16.7–79.9	30.7 (15–57)	14 ± 9.5–26.8
Δ UPDRS (%)	11.2 (0–63.6)	15.6 ± 13–19.3	13.3 (0–53.5)	17.1 ± 12–30.1	20.5 (0–63.6)	21.1 ± 15.3–34.1	3.5 (0–20)	6.4 ± 4.7–10.4	7.6 (0–22.9)	8.3 ± 5.9–13.7
LEDD	731.9 (200–3818)	813.7 ± 618.8–1188.4	1209.3 (250–3818)	1327.2 ± 828.4–3255.1	416.1 (250–863)	229.1 ± 147.7–504.6	717 (200–1368)	561.3 ± 318–2092.7	533.3 (300–1000)	404.1 ± 210.4–2539.9
III ventricle (mm)	7.6 (4.2–12.5)	2.2 ± 1.6–3.4	11.1 (6–16)	2.5 ± 1.8–4	12.2 (5–19)	3.1 ± 2.4–4.6	9.4 (6–14)	2.5 ± 1.9–3.6	10.3 (7–14)	2.1 ± 1.6–3.1
PONS (cm^2^)	4.96 (3.6–6.6)	0.73 ± 0.54–1.13	4.79 (3.75–5.79)	0.61 ± 0.44–1.01	4.77 (4.08–5.55)	0.39 ± 0.29–0.58	4.59 (2.92–5.55)	0.65 ± 0.5–0.94	4.44 (3.32–5.54)	0.61 ± 0.46–0.9
MIDBRAIN (cm^2^)	1.21 (0.7–1.5)	0.23 ± 0.17–0.35	0.76 (0.57–1.03)	0.12 ± 0.09–0.21	0.76 (0.42–1.66)	0.32 ± 0.24–0.47	1.08 (0.76–1.5)	0.19 ± 0.14–0.27	0.81 (0.4–1.27)	0.22 ± 0.17–0.32
M/P ratio	0.245 (0.194–0.313)	0.034 ± 0.025–0.053	0.16 (0.133–0.202)	0.02 ± 0.014–0.033	0.157 (0.098–0.321)	0.056 ± 0.042–0.083	0.239 (0.16–0.344)	0.054 ± 0.041–0.078	0.183 (0.103–0.343)	0.053 ± 0.04–0.079
MRPI	11.247 (7.758–15.429)	1.882 ± 1.39–2.912	17.384 (12.025–26.626)	4.439 ± 3.183–7.327	19.366 (9.343–31.303)	6.944 ± 5.247–10.27	10.766 (7.163–17.151)	2.503 ± 1.915–3.614	15.363 (6.582–29.025)	5.03 ± 3.801–7.439
MRPI 2.0	2.342 (1.252–3.463)	0.619 ± 0.457–0.958	4.338 (2.526–6.306)	1.344 ± 0.964–2.219	5.646 (2.595–9.838)	2.324 ± 1.756–3.436	2.558 (1.41–4.549)	0.91 ± 0.696–1.314	4.195 (1.953–9.285)	1.686 ± 1.274–2.493

Legend: M/F = male/female count; min, minimal value; max, maximal value; SD, standard deviation; CI, confidence interval; UPDRS III on/off, Unified Parkinson Disease Rating Scale (III) on and off stimulation; Δ UPDRS (%), UPDRS difference on and off stimulation in %; LEDD, levodopa equivalent doses; M/P ratio, midbrain to pons ratio; MRPI, magnetic resonance parkinsonian index; MRPI 2.0, magnetic resonance parkinsonian index 2.0. Parameters differentiating PSP-P vs. MSA-P and PSP-P vs. control group are marked in yellow.

**Table 2 jcm-11-02701-t002:** Comparison of PSP-P with PSP-RS, MSA-P and CBS with regard to MRI parameters.

Parameter	PSP-P					
	III Ventricle	PONS (cm^2^)	MIDBRAIN (cm^2^)	M/P Ratio	MRPI	MRPI 2.0
control	0.9215	1.0	0.0002	0.0001	0.0052 *	0.0006
PSP-RS	1.0	1.0	1.0	1.0	1.0	1.0
MSA-P	0.9115	1.0	0.0016	0.0008	0.0003	0.0038 *
CBS	1.0	1.0	1.0	1.0	1.0	1.0

* In terms of multiple comparison correction, we performed a Bonferroni correction to control the false discovery rate (FDR). The calculated *p*-value was 0.002 and was used as a significance threshold for all calculations.

**Table 3 jcm-11-02701-t003:** ROC curve analysis PSP-P vs. MSA-P with regard to midbrain, M/P ratio, MRPI and MRPI 2.0.

Parameter	Comparison	Cut Off	AUC	AUC 95% CI	*p*	Sensitivity	Specificity	Accuracy	PPV	NPV
MIDBRAIN (cm^2^)	PSP-P vs. MSA-P	0.85	0.93	0.846–1.0	0.0000	0.846	0.905	0.882	0.846	0.905
M/P ratio		0.177	0.934	0.856–1.0	0.0000	0.846	0.905	0.882	0.846	0.905
MRPI		12.846	0.817	0.676–0.958	0.0000	0.923	0.619	0.735	0.6	0.929
MIDBRAIN (cm^2^)	PSP-P vs. Control	1.03	0.938	0.849–1	0.0000	1	0.813	0.897	0.813	1
M/P ratio		0.202	0.995	0.98–1	0.0000	1	0.938	0.966	0.929	1
MRPI		12.846	0.928	0.833–1	0.0000	0.923	0.875	0.897	0.857	0.933
MRPI 2.0		3.062	0.971	0.917–1	0.0000	1	0.875	0.931	0.867	1

## Data Availability

The data presented in this study are available on request.

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
