# Peer review of "Is MRPI 2.0 More Useful than MRPI and M/P Ratio in Differential Diagnosis of PSP-P with Other Atypical Parkinsonisms?"

_jcm, 2022, doi:10.3390/jcm11102701_

Round 1
Reviewer 1 Report
In the present study, the authors investigated the utility of MRI parameters for the evaluation of the mesencephalon in the differential diagnosis between parkinsonism predominant type of progressive supranuclear palsy (PSP-P) and PSP-Richardson Syndrome (PSP-RS), corticobasal syndrome (CBS), and multiple sclerosis atrophy - parkinsonian type (MSA-P). An healthy control group was also included. Overall, the study enrolled a total of 74 participants. The MRI parameters included the Magnetic Resonance Parkinsonism Index (MRPI), the MRPI 2.0. and the Mesencephalon/Pons (M/P) ratio. The authors found that only the M/R ratio and MRPI show a significant difference between PSP-P and MSA-P patients. None of the parameters was able to differentiate PSP-P from other atypical tauopathic syndromes.
The study is of interest and the paper is well written. The methods are sufficiently accurate. However, there are some issues, mainly in the results and discussion:
Major comments
- Participants: some important data are missing, including participants’ demographic and clinical features. Please provide information about the gender distribution. Also, please add some clinical information in patients, e.g., the main scores from the most used clinical scales, including the Hoehn and Yahr (H&Y), the Movement Disorder Society-sponsored revision of the Unified Parkinson's Disease Rating Scale (MDS-UPDRS) (Goetz et al., 2008), the PSP Rating Scale (Golbe and Ohman-Strickland 2007) and the Unified MSA Rating Scale (Wenning et al. 2004). Also, if available, provide the levodopa equivalent daily doses in patients.
- Results: tables should be improved. First, tables’ legends are uncompleted. Please indicate all the abbreviations used in the tables. Also, for Table 1, please clarify which data are underlined in yellow. In Table 2, please indicate all the p values, including those statistically insignificant. Finally, it might be useful to graphically show the ROC curves by adding one or two figures instead of entering the data in Table 3.
- The Discussion is relatively poor and several issues need improvement:
- Although the discussion of previous findings in Parkinson's disease (PD) may be useful, it should not have a central role, considering that this condition was not studied in the present work. Instead, the authors should emphasize previous literature findings in the conditions that were included in their work, such as the various forms of PSP, MSA, and CBD. In this regard, some recent important papers have been not mentioned (see for example Quattrone et al., Radiology, 2008, which included PSP, MSA, and PD patients). Also, I found a few inaccuracies. For example, in the papers by Walter et al., 2007 and by Barsottini 2007, the authors found that the third-ventricle dilatation helped in distinguish PSP from PD, and not PSP from other atypical parkinsonisms.
- The authors could discuss the lack of differences between PSP-P and PSP-RS, also in light of previous evidence showing that midbrain atrophy may not be evident in all PSP subtypes and thus, atrophy patterns from other brain regions (such as the globus pallidus, frontal lobe, and cerebral peduncle) may provide additional information (Sakurai et al., Neuroradiology 2017).
- Line 130: ‘Various works stressed the 130 significance of differentiation between PSP and CBS using an assessment of the width of the third ventricle.’ Please specify which works.
Minor comments
Please pay attention to the abbreviations in the Abstract (PSP and M/R should be abbreviated when they appear for the first time), as well as in the main text (see for example page 2, line 47: CBS).
The statement on Page 1, line 37 :’PSP-P is an entity in which symptoms…be present’ is convoluted and not easily understandable. Please rephrase it.
Relevant reference not listed:
Quattrone A, Nicoletti G, Messina D, Fera F, Condino F, Pugliese P, Lanza P, Barone P, Morgante L, Zappia M, Aguglia U, Gallo O. MR imaging index for differentiation of progressive supranuclear palsy from Parkinson disease and the Parkinson variant of multiple system atrophy. Radiology. 2008 Jan;246(1):214-21. doi: 10.1148/radiol.2453061703. Epub 2007 Nov 8. PMID: 17991785.
Sakurai K, Tokumaru AM, Shimoji K, Murayama S, Kanemaru K, Morimoto S, Aiba I, Nakagawa M, Ozawa Y, Shimohira M, Matsukawa N, Hashizume Y, Shibamoto Y. Beyond the midbrain atrophy: wide spectrum of structural MRI finding in cases of pathologically proven progressive supranuclear palsy. Neuroradiology. 2017 May;59(5):431-443. doi: 10.1007/s00234-017-1812-4. Epub 2017 Apr 6. PMID: 28386688.
Author Response
Dear Reviewer 1,
We are grateful for your valuable comments. Please find below revised version of the manuscript and our responses to your suggestions.
- Participants: some important data are missing, including participants’ demographic and clinical features. Please provide information about the gender distribution. Also, please add some clinical information in patients, e.g., the main scores from the most used clinical scales, including the Hoehn and Yahr (H&Y), the Movement Disorder Society-sponsored revision of the Unified Parkinson's Disease Rating Scale (MDS-UPDRS) (Goetz et al., 2008), the PSP Rating Scale (Golbe and Ohman-Strickland 2007) and the Unified MSA Rating Scale (Wenning et al. 2004). Also, if available, provide the levodopa equivalent daily doses in patients.
Data concerning participants’ demographic and clinical features and gender distribution were added. Information describing patients’ clinical state with the use of MDS-UPDRS and LEDDs were implemented. In accordance with the evaluation of clinical state, only MDS-UPDRS part III was used. This assessment was conducted on purpose of facilitating comparison between groups as well as evaluation of levodopa responsiveness among study participants.
- Results: tables should be improved. First, tables’ legends are uncompleted. Please indicate all the abbreviations used in the tables. Also, for Table 1, please clarify which data are underlined in yellow. In Table 2, please indicate all the p values, including those statistically insignificant. Finally, it might be useful to graphically show the ROC curves by adding one or two figures instead of entering the data in Table 3.
Tables were improved in accordance with suggested changes.
- The Discussion is relatively poor and several issues need improvement:
- a) Although the discussion of previous findings in Parkinson's disease (PD) may be useful, it should not have a central role, considering that this condition was not studied in the present work. Instead, the authors should emphasize previous literature findings in the conditions that were included in their work, such as the various forms of PSP, MSA, and CBD. In this regard, some recent important papers have been not mentioned (see for example Quattrone et al., Radiology, 2008, which included PSP, MSA, and PD patients).
The changes were implemented, suggested references were added.
- b) Also, I found a few inaccuracies. For example, in the papers by Walter et al., 2007 and by Barsottini 2007, the authors found that the third-ventricle dilatation helped in distinguish PSP from PD, and not PSP from other atypical parkinsonisms.
The sentences were rephrased in order to highlight the differentiating potential between PSP and PD.
The fact that the parameter which additionally takes into account the width of the third ventricle was not significant is intriguing considering that in multiple studies the width of the third ventricle was found to be a helpful feature in differentiating PSP from PD . Various works stressed the significance of differentiation between PSP and CBS using an assessment of the width of the third ventricle in transcranial sonography. The assessment of the third ventricle was found to be a feature of evolution of PSP.
The references were changed to avoid inaccuracies.
- c) The authors could discuss the lack of differences between PSP-P and PSP-RS, also in light of previous evidence showing that midbrain atrophy may not be evident in all PSP subtypes and thus, atrophy patterns from other brain regions (such as the globus pallidus, frontal lobe, and cerebral peduncle) may provide additional information (Sakurai et al., Neuroradiology 2017).
The information and references were implemented.
- d) Line 130: ‘Various works stressed the 130 significance of differentiation between PSP and CBS using an assessment of the width of the third ventricle.’ Please specify which works.
References were implemented.
Minor comments
Please pay attention to the abbreviations in the Abstract (PSP and M/R should be abbreviated when they appear for the first time), as well as in the main text (see for example page 2, line 47: CBS).
The correction regarding abbreviations was implemented in line 47 and other parts where it was necessary.
The statement on Page 1, line 37 :’PSP-P is an entity in which symptoms…be present’ is convoluted and not easily understandable. Please rephrase it.
The sentence was rephrased:
„PSP-P is a clinical entity with various symptoms overlapping other atypical parkisonisms. Possible dysautonomia and less pronounced cognitive deterioration are examples of symptoms commonly associated with Multiple System Atrophy – Parkinsonian type (MSA-P), however possibly present in PSP-P [2,3]”
Relevant reference not listed:
Quattrone A, Nicoletti G, Messina D, Fera F, Condino F, Pugliese P, Lanza P, Barone P, Morgante L, Zappia M, Aguglia U, Gallo O. MR imaging index for differentiation of progressive supranuclear palsy from Parkinson disease and the Parkinson variant of multiple system atrophy. Radiology. 2008 Jan;246(1):214-21. doi: 10.1148/radiol.2453061703. Epub 2007 Nov 8. PMID: 17991785.
Sakurai K, Tokumaru AM, Shimoji K, Murayama S, Kanemaru K, Morimoto S, Aiba I, Nakagawa M, Ozawa Y, Shimohira M, Matsukawa N, Hashizume Y, Shibamoto Y. Beyond the midbrain atrophy: wide spectrum of structural MRI finding in cases of pathologically proven progressive supranuclear palsy. Neuroradiology. 2017 May;59(5):431-443. doi: 10.1007/s00234-017-1812-4. Epub 2017 Apr 6. PMID: 28386688.
References were added.
Reviewer 2 Report
1. That is an interesting paper attempting to define further imaging parameters in the differentiation of PSP-P from other atypical tauopathic syndromes.
I suggest to explain to the reader less familiar with the neuroradiologic parameters MRPI 2.0, MRPI and M/P ratio and to provide images to visualize the differences in these parameters.
2. The control group needs further detailing as to gender, co-morbidities.
Author Response
Dear Reviewer 2,
We are grateful for your valuable comments. Please find below modified version of the manuscript and our responses to your suggestions.
- I suggest to explain to the reader less familiar with the neuroradiologic parameters MRPI 2.0, MRPI and M/P ratio and to provide images to visualize the differences in these parameters.
Methods of calculating radiological parameters mentioned in the manuscript were explained in detail. Images visualizing the parameters were added.
- The control group needs further detailing as to gender, co-morbidities.
Additional information was added. Authors also implemented the reference to Table 1 in the material section. In the context of comorbidities authors added a statement:
Except for hypertension, headaches and epilepsy no clinically significant comorbidities were observed in the control group.
Round 2
Reviewer 1 Report
Comments to the authors:
The authors have appropriately addressed my comments and made meaningful changes in the manuscript.
I have only few minor addition comments:
- The white lines in the MRI figures are not clearly visible (at least in the PDF). Please increase the thickness or try to enhance the contrast between the lines and the images. Also, figure legends could be improved. Please clarify whether the images are extracted from an MRI of a healthy subject or a patient. Please, add some details concerning the calculation of the various parameters.
- Line 36: please replace ‘this entity’ with PSP-P.
- Line 38: the sentence ‘PSP-P is a clinical entity with various symptoms overlapping other atypical parkinsonisms’ is redundant. Please delete it.
- The reference to the images 6-9 should be made in the Result section (page 7) instead of the Methods sections.
Author Response
Dear Reviewer 1,
Thank you for your valuable suggestions. Please find below our responses to your comments.
- The white lines in the MRI figures are not clearly visible (at least in the PDF). Please increase the thickness or try to enhance the contrast between the lines and the images. Also, figure legends could be improved. Please clarify whether the images are extracted from an MRI of a healthy subject or a patient. Please, add some details concerning the calculation of the various parameters.
MRI images were changed in order to increase its clarity. Figure legends were improved. Details describing the calculation of different parameters were presented in subsection 2.1. - Line 36: please replace ‘this entity’ with PSP-P.
Change implemented. - Line 38: the sentence ‘PSP-P is a clinical entity with various symptoms overlapping other atypical parkinsonisms’ is redundant. Please delete it.
Change implemented. - The reference to the images 6-9 should be made in the Result section (page 7) instead of the Methods sections.
Change implemented.
Reviewer 2 Report
In the text you have crossed out sentences and re-written exactly the same sentences. Idem in the reference section: references crossed out and the same re-indicated. On the first glance it appears that the manuscript was revised in detail however that does not seem so. Also in the images one can hardly see what was measured. Please use colors and arrows to clearly indicate to the reader what was measured.
Please detail more your introduction. Why imaging is relevant in PSP-P. You mention it in the discussion. But already the introduction should "introduce" the reader in this specific topic.
Author Response
Dear Reviewer 2,
We are grateful for your attentive comments. Please find below our responses.
In the text you have crossed out sentences and re-written exactly the same sentences. Idem in the reference section: references crossed out and the same re-indicated. On the first glance it appears that the manuscript was revised in detail however that does not seem so.
Authors would like to sincerely apologize for the inconvenience regarding the “follow changes mode”. The parts of the text, which were highlighted in red are the ones that were added in accordance with the changes requested by the Reviewers from the first round. The changes may look awkward due to the fact, that primarily authors highlighted the changes in red and in the aftermath, due to the policy of the journal, changed the modifications in red to “follow changes mode”. In the current version newly added fragments or changes (compared to the first version of the manuscript) are marked in red without the "follow changes mode" in order to increase the clarity of the manuscript. Once again, we sincerely apologize for inconvenience regarding unsuitable editing.
Also in the images one can hardly see what was measured. Please use colors and arrows to clearly indicate to the reader what was measured.
Images were improved.
Please detail more your introduction. Why imaging is relevant in PSP-P. You mention it in the discussion. But already the introduction should "introduce" the reader in this specific topic.
The following part was implemented to the introduction: "In PSP-P the factors assessed in neuroimaging remain not fully explored. The MRI parameters are commonly use to perform differential diagnosis of PSP-P and PD in the early stages, less is known in the context of comparative analysis of PSP-P and MSA [5,6]. Among the features commonly associated with PSP-P are the evaluation of Magnetic Resonance Parkinsonism Index (MRPI) and its upgradet edition (MRPI 2.0)."